

# The SINDRUM-I experiment

**Ralph Eichler and Christoph Grab⋆**

Department of Physics, ETH Zürich, 8093 Zürich, Switzerland

⋆ grab@ethz.ch

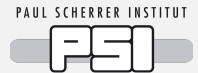

## Abstract

SINDRUM-I was the first nearly $4\pi$ spectrometer at SIN. It was initially designed to search for the forbidden decay $\mu^+ \to e^+e^-e^+$, but also successfully studied various other processes with high precision. The upper limit obtained for the branching ratio of $B_{\mu\to3e} = \Gamma(\mu^+ \to e^+e^-e^+)/\Gamma(\mu^+ \to e^+\nu_e\bar{\nu}_\mu) < 1.0 \times 10^{-12}$ (90% CL) from 1988 is still the best. The first statistically significant observation of the rare decay $\mu^+ \to e^+e^-e^+\nu_e\bar{\nu}_\mu$, achieved in 1985, yielded a branching ratio of $B_{\mu\to3e2\nu} = (3.4 \pm 0.2 \pm 0.2) \times 10^{-5}$. Several other measurements of rare processes were undertaken. The first observation of the $\pi$-decay $\pi^+ \to e^+\nu_e e^-e^+$ resulted in the value $\Gamma(\pi^+ \to e^+\nu_e e^-e^+)/\Gamma(\pi^+ \to \mu^+\nu_\mu) = (3.2 \pm 0.5 \pm 0.2) \times 10^{-9}$, also still the best measurement. The determination of the ratio of the weak axial- to vector-form factor $F_A/F_V = (0.7 \pm 0.5)$ resolved a long-standing ambiguity. In addition, upper limits for $\mu^+ \to e^+\phi$ and $\pi^+ \to e^+\nu_e\phi$ with subsequent decay $\phi \to e^+e^-$ (search for "massless" Goldstone bosons $\phi$) and $\pi^0 \to e^+e^- < 1.3 \times 10^{-7}$ were obtained.

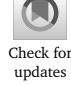

## 7.1 History - how it all began

In the fall of 1976 rumors spread about an experiment performed at SIN for the search of the decay $\mu \to e\gamma$. A debate was going on, whether or not the decay had been observed. The rumors traveled from SIN via email to R. Eichler at Stanford and from him to a graduate student in the lecture-class of James Bjorken. The next week, J. Bjorken in turn gave the students an exercise to compute the decay rate and also confronted his colleague Steven Weinberg with the rumor. It took a few weeks after Weinberg's talk at the APS meeting to reach the New York Times. There it read on February $8^{th}$ 1977: *Experimenters in Switzerland have reportedly observed an "impossible" transmutation of atomic particles. This has thrown the world community of theoretical physicists into a frenzy of speculations, calculations and publications (S. Weinberg).* This inspired R. Hofstadter of Stanford to initiate an experiment at LAMPF for $\mu^+ \to e^+\gamma$ to try to resolve the dispute around the SIN experiment.

The results from the SIN experiment were finally published as an upper limit for the muon decay $\mu \to e\gamma$. However, all these speculations triggered a wider range of searches of muon flavour violating decays at LAMPF and SIN, and these activities continue presently at PSI, Fermilab and J-PARC.

## 7.2 The lepton flavour violating process $\mu^+ \to e^+e^-e^+$

In the Standard Model (SM), charged lepton flavour violating reactions (LFV) are forbidden at tree level and can only be induced by lepton mixing through higher-order diagrams. One of the dominant contributions, the mixing through loop diagrams with massive neutrinos, see Figure 7.1a, is strongly suppressed in the SM with a predicted branching ratio B below the level of $10^{-50}$ [1] . Thus, the decay $\mu^+ \to e^+e^-e^+$ potentially provides very high sensitivity to LFV reactions in various models of physics Beyond the Standard Model, in which the couplings are mediated by completely new particles.

At the time of the SINDRUM-I experiment, lepton flavour violation in the neutral lepton sector (neutrino oscillations) were not yet established, and theories were focused on extensions of the SM by introducing different new heavy particles that can mediate charged LFV either in virtual loops (Figure 7.1b), at tree level (see Figure 7.1c), or in box diagrams. These new models included right-handed bosons, additional Higgs doublets, neutral scalar singlets, familons, extended technicolor gauge bosons, doubly charged so-called "heptons", various "horizontal" models, and notably supersymmetric (SUSY) models with scalar leptons. An example is Figure 7.1b, in which a $\gamma/Z$-penguin diagram is shown with new SUSY particles running in a loop. These loop contributions are important for all models where new particle couplings to electrons and muons are introduced.

Not all of these models have survived with equal popularity today. However, modern models also include new particles such as Higgs particles or doubly charged Higgs particles, R-parity-violating scalar neutrinos, supersymmetric particles and new heavy vector bosons.

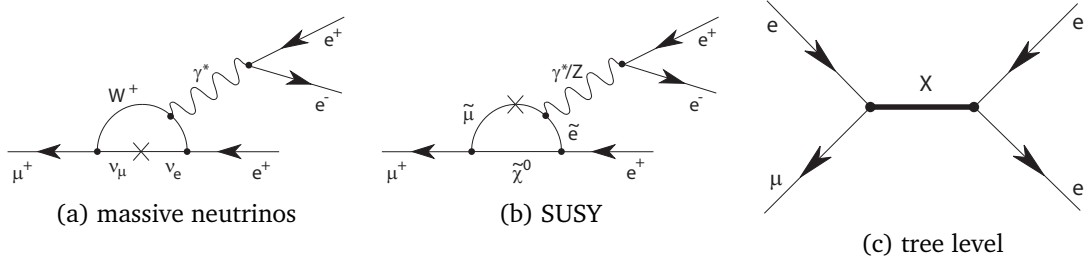

(a) massive neutrinos      (b) SUSY      (c) tree level

Figure 7.1: Feynman diagrams for lepton flavour violation in $\mu^+ \to e^+e^-e^+$. (a) by massive neutrino mixing; (b) by heavy mediating particles, such as in SUSY models; (c) tree level mediating particles.

## 7.3 What physics did we learn from the SINDRUM-I experiment ?

### 7.3.1 Search for the decay $\mu^+ \to e^+e^-e^+$

The main focus of the SINDRUM I experiment was the search for the decay $\mu^+ \to e^+e^-e^+$ [2–4], with the aim to improve the sensitivity substantially beyond the then existing limits of $B_{\mu\to3e} < 1.9 \times 10^{-9}$ at 90% C.L. [5].

The unique kinematic topology of the 3-body decay was exploited in the analysis, namely three identical-mass electrons (and positrons) with all tracks originating from one common vertex, coincident in time, with vanishing total momentum and a total energy equal to the

muon mass. The dominant background stems from accidental combinations of tracks (e.g. in combination with Bhabha scattering) and from the irreducible, allowed but strongly suppressed internal radiative decay $\mu^+ \to e^+ e^- e^+ \nu_e \bar{\nu}_\mu$. The data reduction was achieved with a multiple stage trigger, taking advantage of track and charge preselectors, requiring at least one negatively and two positively charged tracks within a time window of 7 ns. This was complemented by a track correlator which limited the total transverse momentum of the $e^+ e^- e^+$-triplet to below 17 MeV/$c$. A full three-dimensional event reconstruction was performed offline. As an example, a reconstructed $\mu^+ \to e^+ e^- e^+$ event candidate is shown in Figure 7.2b. The acceptances and efficiencies were determined by Monte Carlo simulations. Prompt events were distinguished from accidentals by time difference constraints between the mean time of the $e^+ e^-$-pair and the time of the second $e^+$. The final number of potentially observed $\mu^+ \to e^+ e^- e^+$ candidate decays was determined from the 2-dimensional distribution of ($\sum E_i$ vs $\hat{p}^2$) for both the prompt and the accidental events. Energy conservation requires $\sum E_i = m_\mu$ within errors for true $\mu^+ \to e^+ e^- e^+$ events, and $\hat{p}^2 = (p_\parallel / \sigma_{p_\parallel})^2 + (p_\perp / \sigma_{p_\perp})^2$ to be centered at zero. The distribution is shown in Figure 7.2a for the measured prompt events. No events were observed within the indicated 95% C.L. contour for $\mu^+ \to e^+ e^- e^+$ decays. Based on zero observed events an upper limit on the decay branching ratio $B_{\mu^+ \to e^+ e^- e^+}$ was determined by normalising to the number of observed $\mu^+ \to e^+ e^- e^+ \nu_e \bar{\nu}_\mu$ events. Already during construction of SINDRUM-I with four out of five tracking chambers an order of magnitude better limit [2] compared to [5] was published. Combining then the data from all running periods, the final branching ratio obtained [4] was

$$B_{\mu \to 3e} < 1.0 \times 10^{-12} \quad \text{at 90\% C.L..} \tag{7.1}$$

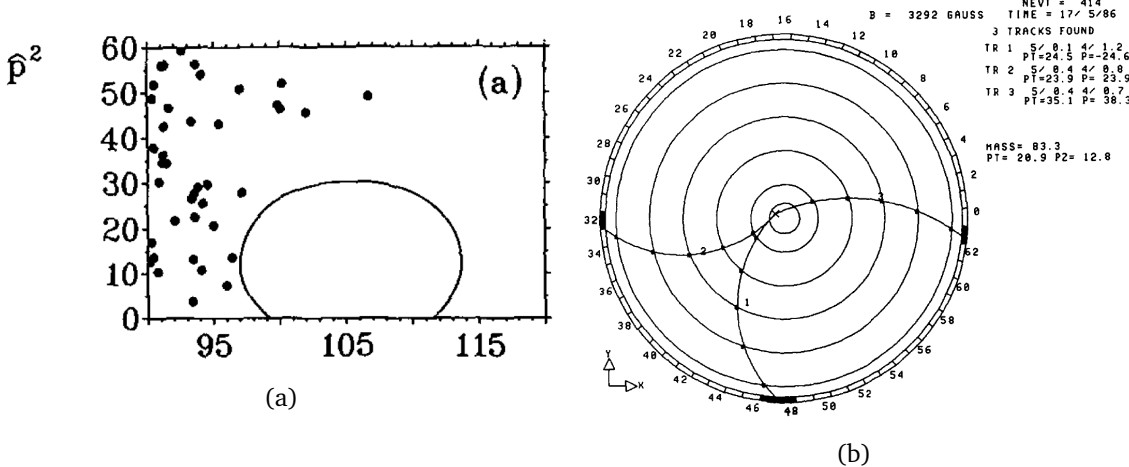

(a)

(b)

Figure 7.2: (a) Distribution of the ($\Sigma E_i$ vs $\hat{p}^2$) for prompt events; the contour defines the 95% C.L. region for $\mu^+ \to e^+ e^- e^+$ decays. (b) Example of a reconstructed $\mu^+ \to e^+ e^- e^+$ candidate event, shown in the $r - \phi$ plane.

### 7.3.2 Measurement of the internal radiative decay $\mu^+ \to e^+ e^- e^+ \nu_e \bar{\nu}_\mu$

The internal radiative decay $\mu^+ \to e^+ e^- e^+ \nu_e \bar{\nu}_\mu$ constitutes the main irreducible background contribution for the $\mu^+ \to e^+ e^- e^+$ search. This rare decay is also of interest itself as it can be calculated to a precision below the per mille level. Hence, this decay was also analysed in parallel to $\mu^+ \to e^+ e^- e^+$, using the same time and vertex constraints. During the first data taking runs with the complete SINDRUM-I detector, a total of $N = (7.3 \pm 0.5) \cdot 10^{12}$

muons were stopped in the target and were used for the analyses of both $\mu^+ \to e^+ e^- e^+$ and $\mu^+ \to e^+ e^- e^+ \nu_e \bar{\nu}_\mu$. Based on the observation of 7443 $\mu^+ \to e^+ e^- e^+ \nu_e \bar{\nu}_\mu$ events and an efficiency of $3 \times 10^{-5}$, a decay branching ratio of $B_{\mu \to 3e2\nu} = (3.4 \pm 0.2 \pm 0.2) \times 10^{-5}$ was measured [3], consistent with the SM prediction, and is still the most accurate as of this writing. Previous experiments had only been able to observe a handful of events ($\leq 7$ events). Thus, this was the first statistically significant observation of the $\mu^+ \to e^+ e^- e^+ \nu_e \bar{\nu}_\mu$ decay.

### 7.3.3 Measurement of $\pi^+ \to e^+ \nu_e e^- e^+$

In the decays $\pi^+ \to e^+ \nu_e \gamma$ and $\pi^+ \to e^+ \nu_e e^- e^+$, both the vector- and axial-vector weak hadronic currents contribute to the decay amplitudes and are parameterized by the vector and axial vector form factors $F_V$ and $F_A$, respectively. There is a firm prediction for the value of $F_V$. The conserved vector current rule connects $F_V$ with the $\pi^0$ lifetime so that $|F_V| = 0.0255$, but the sign is undetermined. Contrary to the case of $\pi^+ \to e^+ \nu_e \gamma$, the ratio of $F_A/F_V$ is unambiguously measurable in the decay $\pi^+ \to e^+ \nu_e e^- e^+$ and the result of [6] excludes a possible negative value of $F_A/F_V$ from the $\pi^+ \to e^+ \nu_e \gamma$ experiments. In the high statistics run of SINDRUM-I [7] the first determination of

$$B_{\pi^+ \to e^+ \nu_e e^- e^+} = \Gamma(\pi^+ \to e^+ \nu_e e^- e^+)/\Gamma(\pi^+ \to \mu^+ \nu_\mu) = (3.2 \pm 0.5 \pm 0.2) \times 10^{-9} \qquad (7.2)$$

was achieved, where the first error is the statistical uncertainty and the second error is due to the uncertainty of the form factors. This $B_{\pi^+ \to e^+ \nu_e e^- e^+}$ still holds as of this writing. By fixing the value $F_V = 0.0255$ the form factor $F_A = 0.019 \pm 0.008$ was determined.

### 7.3.4 Search for light particles produced in muon- or pion decays

Many theories beyond the Standard Model predict "massless" Nambu-Goldstone bosons arising from the breaking of an underlying symmetry. Examples are the "familon" for a broken family hierarchy, the "axion" for a broken axial baryon number proposed to solve the strong CP problem, the majoron, and neutral scalar bosons.

In the search for a light Higgs $h$ in the decay $\pi^+ \to e^+ \nu_e h$, where the Higgs decays in $h \to e^+ e^-$, the same selection criteria as for the analysis of the pion form factors were applied [7]. Higgs particles with a decay length less than the vertex resolution of the SINDRUM detector should be visible in the decay $\pi^+ \to e^+ \nu_e e^- e^+$ as a peak in the $e^+ e^-$ -invariant mass distribution. No such signal was observed for Higgs masses $2m_e < m_h < 110$ MeV/$c^2$.

A similar search was made for an axion-like neutral particle produced in both $\mu$ or $\pi$ decays, $\mu^+ \to e^+ \phi$ and $\pi^+ \to e^+ \nu \phi$, with a subsequent decay $\phi \to e^+ e^-$. No candidates were found, and therefore upper limits for the branching ratios were determined as a function of the $\phi$ masses and lifetimes. For $\phi$ lifetimes below $10^{-10}$ s limits on $B$ down to $2 \times 10^{-12}$ were obtained [8].

Furthermore, a search for weakly interacting neutral bosons (X) produced in $\pi^- p$ interactions at rest and decaying into $e^+ e^-$ pairs was performed with the SINDRUM detector. The data sample searched contained 98400 $\pi^0 \to e^+ e^- \gamma$ decays and 27200 $\pi^- p \to n e^+ e^-$ events, each with an $e^+ e^-$ invariant mass between 25 and 139 MeV/$c$. Upper limits for the branching ratios $\Gamma(\pi^0 \to X\gamma, X \to e^+ e^-)/\Gamma(\pi^0 \to all)$ and $\Gamma(\pi^- p \to Xn, X \to e^+ e^-)/\Gamma(\pi^- \to all)$ for X lifetimes between $10^{-23}$s and $10^{-11}$s were obtained. Upper limits at 90% C.L. range from $10^{-3}$ at an invariant $e^+ e^-$ mass of 25 MeV/$c^2$ to $10^{-5}$ at 100 MeV/$c^2$ [9].

### 7.3.5 Measurement of the decay $\pi^0 \to e^+ e^-$ and $\pi^0 \to e^+ e^- \gamma$

The large helicity suppression of the electromagnetic amplitude of the decay $\pi^0 \to e^+ e^-$ has led to speculations that additional contributions might be important. Anomalous quark-lepton

couplings could lead to significant enhancements of the value for this branching ratio. A branching ratio above the unitarity value would be a sign of CP violating neutral currents. The reaction $\pi^- p \rightarrow \pi^0 n$ at rest was used as a source of tagged mono - energetic $\pi^0$ in a search for the decay $\pi^0 \rightarrow e^+ e^-$ with the SINDRUM I spectrometer. The measurement resulted in [10]

$$B_{\pi^0 \rightarrow e^+ e^-} = \Gamma(\pi^0 \rightarrow e^+ e^-)/\Gamma(\pi^0 \rightarrow \gamma\gamma) < 1.3 \times 10^{-7} \text{ at } 90\% \text{ C.L.,} \tag{7.3}$$

consistent with the QED prediction $B_{\pi^0 \rightarrow e^+ e^-} = (6.5 \pm 0.5) \times 10^{-8}$. The combined result of two previous measurements, $B_{\pi^0 \rightarrow e^+ e^-} = (1.8 \pm 0.7) \times 10^{-7}$, had suggested sizeable additional contributions to the decay amplitude. This possibility seemed most likely ruled out by the SINDRUM result.

In the decay $\pi^0 \rightarrow e^+ e^- \gamma$, the hadronic structure of the pion is parameterized by a form factor $F = 1/(1 - ax)$ with $x = m_{e^+ e^-}/m_{\pi^0}$. The SINDRUM-I analysis of the Dalitz plot distribution measured the value as $a = 0.02 \pm 0.02 \pm 0.04$ [11] with the uncertainties being statistical and systematic, respectively. This value is consistent with the prediction of vector meson dominance of $a \approx 0.03$.

## 7.4 General description of the SINDRUM-I Apparatus

A schematic view of the SINDRUM spectrometer is given in Figure 7.3, with the coordinate system shown. With the help of the evacuated solenoid S, a surface muon beam with momentum 25 MeV/$c$ and intensity $7 \times 10^6 \text{ s}^{-1}$ (produced by a 120 $\mu A$ proton current extracted from the cyclotron) was refocussed from the entrance collimator to the target T, where it stopped. The target was a hollow double-cone shaped body of 58 mm diameter and 220 mm length made of Rohacell[1] with a thickness of 1 mm (11 mg/cm$^2$). The cylindrical magnet with a normal conducting coil M produced a homogeneous ($\Delta B/B < 1\%$) magnetic field of up to 0.6 T parallel to the symmetry axis ($z$-axis) in a volume of 110 cm length × 75 cm diameter. Tracks of decay particles were measured with five concentric self-supporting cylindrical multiwire proportional chambers C of low mass density. Three of them were equipped with cathode strips in order to obtain $z$-coordinates for three-dimensional reconstruction of tracks. For a field of $B = 0.334$ T, as used in the experiment, the momentum resolution is $\Delta p/p = (12.0 \pm 0.5)\%$ and $(8.5 \pm 0.5)\%$ (FWHM) for p = 50 MeV/$c$ and 20 MeV/$c$, respectively. The angular resolution at the target is $\Delta\theta = (65 \pm 3)$ mrad (FWHM) for tracks of 20 MeV/$c$ momentum. Fast timing signals were obtained from the cylindrical scintillator hodoscope H placed between the coil M and the chambers C. The 64 hodoscope elements were viewed at both ends by photomultipliers P. A time resolution of $\Delta t = 0.57$ ns (FWHM) between two hodoscope counters was obtained after correcting for walk and time of flight. The solid angle covered by the spectrometer was 0.73 of $4\pi$.

## 7.5 The low mass multiwire proportional chamber (MWPC)

A main issue of concern for the design of SINDRUM was multiple scattering of the low-energy electrons. A very low mass for the target and the tracking chambers was a real challenge. The spectrometer was equipped with five very thin cylindrical MWPCs, three of which had cathode strip readouts. Each chamber consisted of two concentric Kapton/Rohacell sandwich cylinders, which were assembled on steel mandrels. Glass-fiber epoxy rings were glued to the ends of the cylinders supporting printed circuit rings onto which the 20$\mu$m anode wires, resistors, condensors, and multipin connectors were soldered. The cathodes of chambers 1, 3, and 5 consisted of strips of aluminum evaporated on Kapton having an angle of ±45° for the outer and inner cathodes, respectively. The strips were connected to end printed circuit boards

---

[1]Rohacell manufactured by Röhm Gmbh, Darmstadt, Germany

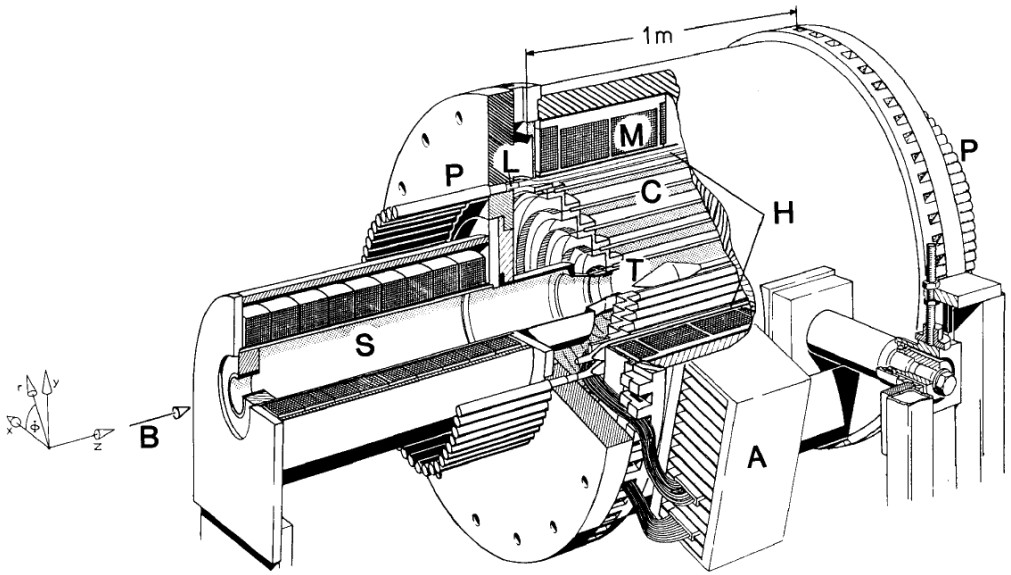

Figure 7.3: The SINDRUM I detector in the horizontal operating orientation.

with conductive paint. The strips of chamber 1 were divided in the middle and read out at both ends of the chamber to reduce the rate per strip. The chambers were operated with a gas mixture of 49.9% Ar, 49.9% $C_2H_6$ and 0.2% freon at a gas gain of $\sim 5 \times 10^4$. The chamber electrodes were connected through 1 m long 75$\Omega$ coaxial cables to the amplifiers mounted around the circumference of the magnet. The spatial resolution of the $\varphi$-measurement was limited by the wire spacing of 2 mm ($\sigma \simeq 0.6$ mm) and the $z$-resolution was determined with cosmic rays to be $\sigma \simeq 0.3$ mm. The chambers were successfully operated throughout the lifetime of the SINDRUM-I experiment. Their conception not only served as an important rôle model for part of the H1-detector construction at the HERA ring in Hamburg, but also laid ground for a very fruitful cooperation between ETH Zurich, Univ. of Zurich and SIN (PSI today).

## 7.6 Summary

The highlight of the SINDRUM-I experiment is clearly the improvement of the sensitivity on the rare decay $\mu^+ \to e^+ e^- e^+$ by three order of magnitudes, reaching an upper limit $BR < 1.0 \times 10^{-12}$ at 90% C.L. The experiment was statistically limited and was not suffering from backgrounds. However, to gain another order of magnitude in precision, a much higher intensity of the muon beam would have been required. Thus, the successor experiment, SINDRUM-II, concentrated on the complementary muon-electron conversion process. As the SINDRUM-I detector resolution was not sufficient for competitive $\mu - e$-conversion measurements, a major upgrade of the detector was done, followed by measurements achieving best upper limits for $\mu - e$-conversion [12, 13].

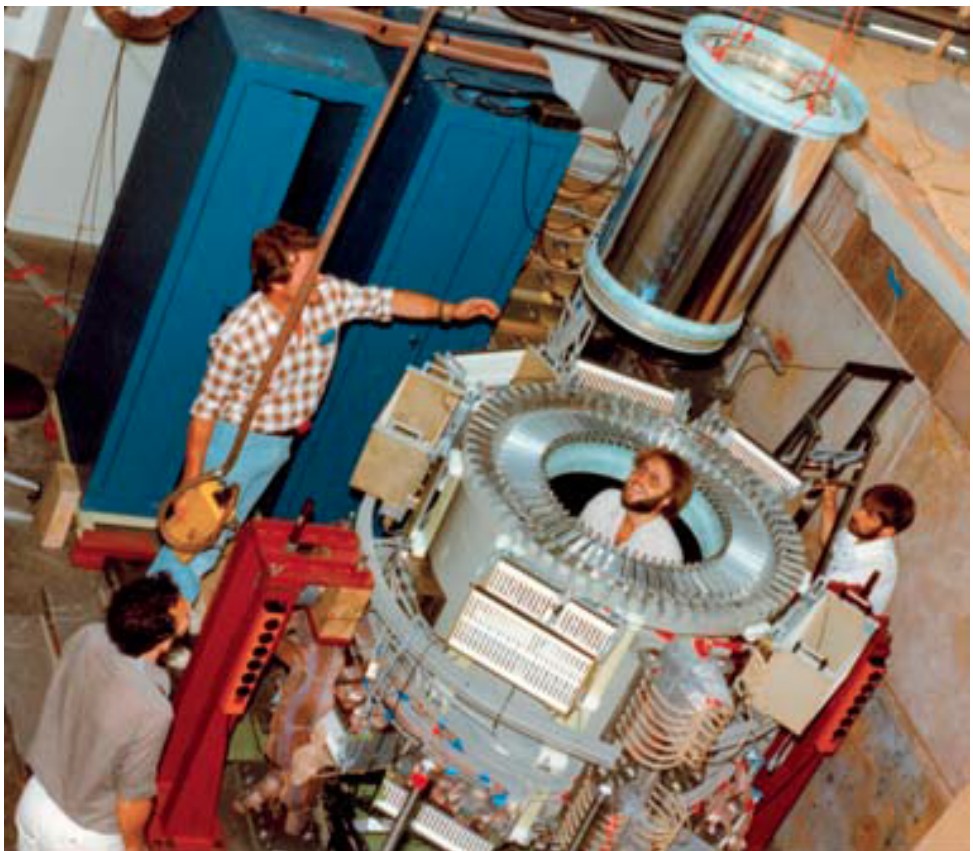

Figure 7.4: The assembly of the SINDRUM I detector in the vertical orientation. The MWPC are being lowered into the setup by (clockwise from top left) Erwin Hermes (technician UZH), Norbert Kraus (PhD student UZH), Nik Lordong (Technician PSI), and within the setup Michael Doser (Master student ETHZ).

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
