# Peer review of "The SINDRUM-I Experiment"

_SciPost Physics, doi:SciPost Phys. Proc. 5, 007 (2021)_

## Round 1 · Referee Report · Niklaus Berger · 2021-1-13

Strengths

1-Provides an inside perspective of the detector and physics achievements of SINDRUM
2-Clearly lists the measurements performed including motivations and results
3-Places the experiment in its historical context

Weaknesses

1-Does not discuss the limitations of the setup and what motivated the end of the experiment
2-Misses a conclusion

Report

The SINDRUM I experiment was an important step in the history of PSI (then SIN) and has performed many measurements which are world leading to this date. The paper clearly outlines the measurements performed and should clearly be publshed as part of a review of particle physics at PSI. The paper is clearly written and very readable. It starts with the historical motivation, introduces the process $\mu \rightarrow eee$ and then describes the measurements performed and their results. The final sections describe the apparatus and in particular the innovative MWPCs used. I am missing a conclusion, which could also discuss the limitations of the appartus/experimental approach and the reasons why the measurement programme was stopped at the point where it was stopped.

Requested changes

1- Line 13: This sentence is a bit long and the form factor ratio does not really fit topically as it is not obviously a rare decay measurement - maybe you can split this into two.
2- Line 18/Lines 113ff.: I find the use of $\phi$ for the new boson confusing - is there a notation with no overlap with mesons?
3- Line 32: I find the jump from activities triggered in 1977 to experiments planned today very abrupt, maybe this can be expanded by a sentence or two (which would then also allow for cross references to other entries in the volume once this is technically possible)
4 - Line 37: Maybe cite the calculations that lead to this number and also mention the state of knowledge when SINDRUM was conceived
5 - Line 39: .. models of physics beyond...
6 - Line 41: How is this different from today?
7 - In several places in the paper you use "still valid today" for results that are world leading/unique to this day. I suggest stronger wording.
8 - Line 102: a verb seems to be missing
9 - Line 143: surface muon beam
10 - Line 149: 110 cm length
11 - Line 152: z should be italic
12 - Line 170: I find the hyphen in end-printed unhelpful
13 - The end of the paper is very abrupt - one might consider swapping the detector and physics parts or - and this would be my preferred solution- add a concluding paragraph briefly sumarizing the highlights and also mentioning the limitations of the experiment, why it was terminated and what came next. (I was always wondering about the exact relationship between SINDRUM-I and -II)

  • validity: top
  • significance: top
  • originality: good
  • clarity: high
  • formatting: excellent
  • grammar: excellent

Author:  Christophorus Grab  on 2021-02-03  [id 1203]

(in reply to Report 1 by Niklaus Berger on 2021-01-13)

“The SINDRUM-I Experiment”

Answers to Referee’s comments and requested changes:

1- Line 13: This sentence is a bit long and the form factor ratio does not really fit topically as it is not obviously a rare decay measurement - maybe you can split this into two. A => RE-phrased

2- Line 18/Lines 113ff.: I find the use of ϕ for the new boson confusing - is there a notation with no overlap with mesons? A => that was the nomenclature customary at that time, and also what was used in the original paper. We decided to keep it.

3- Line 32: I find the jump from activities triggered in 1977 to experiments planned today very abrupt, maybe this can be expanded by a sentence or two (which would then also allow for cross references to other entries in the volume once this is technically possible) A => Added some more info and re-phrased.

4 - Line 37: Maybe cite the calculations that lead to this number and also mention the state of knowledge when SINDRUM was conceived. A => The question on the state of theory is implicitly answered in lines 50ff, where modern models are contrasted with the earlier ones. also added comment about nLFV. Citation added.

5 - Line 39: .. models of physics beyond... A => corrected

6 - Line 41: How is this different from today? A => see added comments in item 4), and further rephrasing.

7 - In several places in the paper you use "still valid today" for results that are world leading/unique to this day. I suggest stronger wording. A => Re-phrased

8 - Line 102: a verb seems to be missing A => corrected

9 - Line 143: surface muon beam A => corrected

10 - Line 149: 110 cm length A => corrected

11 - Line 152: z should be italic A => corrected

12 - Line 170: I find the hyphen in end-printed unhelpful A => corrected

13 - The end of the paper is very abrupt - one might consider swapping the detector and physics parts or - and this would be my preferred solution- add a concluding paragraph briefly summarizing the highlights and also mentioning the limitations of the experiment, why it was terminated and what came next. (I was always wondering about the exact relationship between SINDRUM-I and -II)

A => we added a summary paragraph summarizing and explaning.

---

## Round 1 · Referee Report · Adrian Signer · 2021-1-14

Report

We (the editors Cy Hoffman, Klaus Kirch, Adrian Signer) had the opportunity to review an earlier draft of the article and were in communication with the authors before the submission. All our comments and suggestions have been taken into account. Hence, we think the paper can now be published in the current form.

---

## Editorial Decision

published